# Deformations of Mining Terrain Caused by the Partial Exploitation in the Aspect of Measurements and Numerical Modeling

**Jan Białek, Marek Wesołowski, Ryszard Mielimąka and Paweł Sikora \*** 

Department of Mining, Safety Engineering and Industrial Automation, Faculty of Mining, Silesian University of Technology, Akademicka 2A Street, 44-100 Gliwice, Poland; jan.bialek1@gmail.com (J.B.); marek.wesolowski@polsl.pl (M.W.); ryszard.mielimaka@polsl.pl (R.M.)

\* Correspondence: pawel.sikora@polsl.pl

**Abstract:** The article presents the results of geodetic measurements and numerical modeling of mining area deformations in the partial exploitation area of the $712/_{1-2}$ seam at the Marcel Coal Mine. An important element in this exploitation is the limitation of the length of longwalls with cavings to 130 m and 150 m, leaving an unextracted 70 m wide coal solid belt between them. Leaving the belts aimed to reduce deformations of the mining terrain, with relatively limited deposit losses. The numerical modeling of mining terrain deformations was performed using the Fast Lagrangian Analysis of Continua (FLAC) software package based on the finite difference method. The results of the geodetic measurements and computer simulations presented in the article confirm the assumption adopted during the planning stage of this exploitation about the possible significant reduction of mining terrain deformations caused by leaving the unextracted belts of coal solid between successive longwall panels.

**Keywords:** partial mining exploitation; numerical modeling; terrain deformations; surface protection

## 1. Introduction

In order to achieve good economic results, the Marcel Coal Mine should remain in the group of coking coal producers for as long as possible. These coal types are located in the part of the parent mine, located in the Niedobczyce district of the city of Rybnik, and to a small extent in the northeastern part of the town of Radlin. The subject of the study is the exploited seam $712/_{1-2}$ with a thickness of approximately 3 m, located at a depth of 960 to 1120 m. This is the last seam that can be extracted in this part of the deposit.

The multi-seam mining exploitation conducted to date in the analyzed area of the mine caused the occurrence of significant deformations of the mining terrain surface and mining tremors, which were the cause of protests issued by the residents of that area. Based on the requirements for the protection of surface objects, limit values of deformation rates were established. For most objects in the area, the horizontal deformations valuing $\varepsilon = 3.0$ mm/m and the slope of the subsidence trough profile valuing T = 5.0 mm/m must not be exceeded.

According to the preliminary forecasts [1], the planned roof caving exploitation of the $712/_{1-2}$ seam carried out to the full extent (without leaving belts of coal solid) would cause an emergence exceeding the permissible values of indicators.

This fact incurred the strong resistance of the residents, which led to the decision to abandon mining activities in this area. The only socially acceptable solution was exploitation causing deformations in the range of assumed admissible deformation values for buildings and subsidence in the area of Nacyna river being limited to 0.1 m.

The obvious way of reducing deformations is limiting the height of exploited longwalls, which, apart from losses in the deposit, would contribute to a significant increase in the fire hazard due to leaving large coal resources in the abandoned workings.

This fact prompted the authors [1] to develop a project of seam $712/_{1-2}$ exploitation, which, providing the maximum utilization of the deposit, would lead to the reduction of static influences and significantly limit the frequency and energy of possible tremors. The project, adopted for implementation in 2012 (Figure 1), assumed that seam $712/_{1-2}$ would be mined by means of five longwalls with lengths of 130 m (longwalls M-4 and M-5) and 150 m (longwalls M-1, M-2 and M-3), leaving 70 m wide pillars of coal.

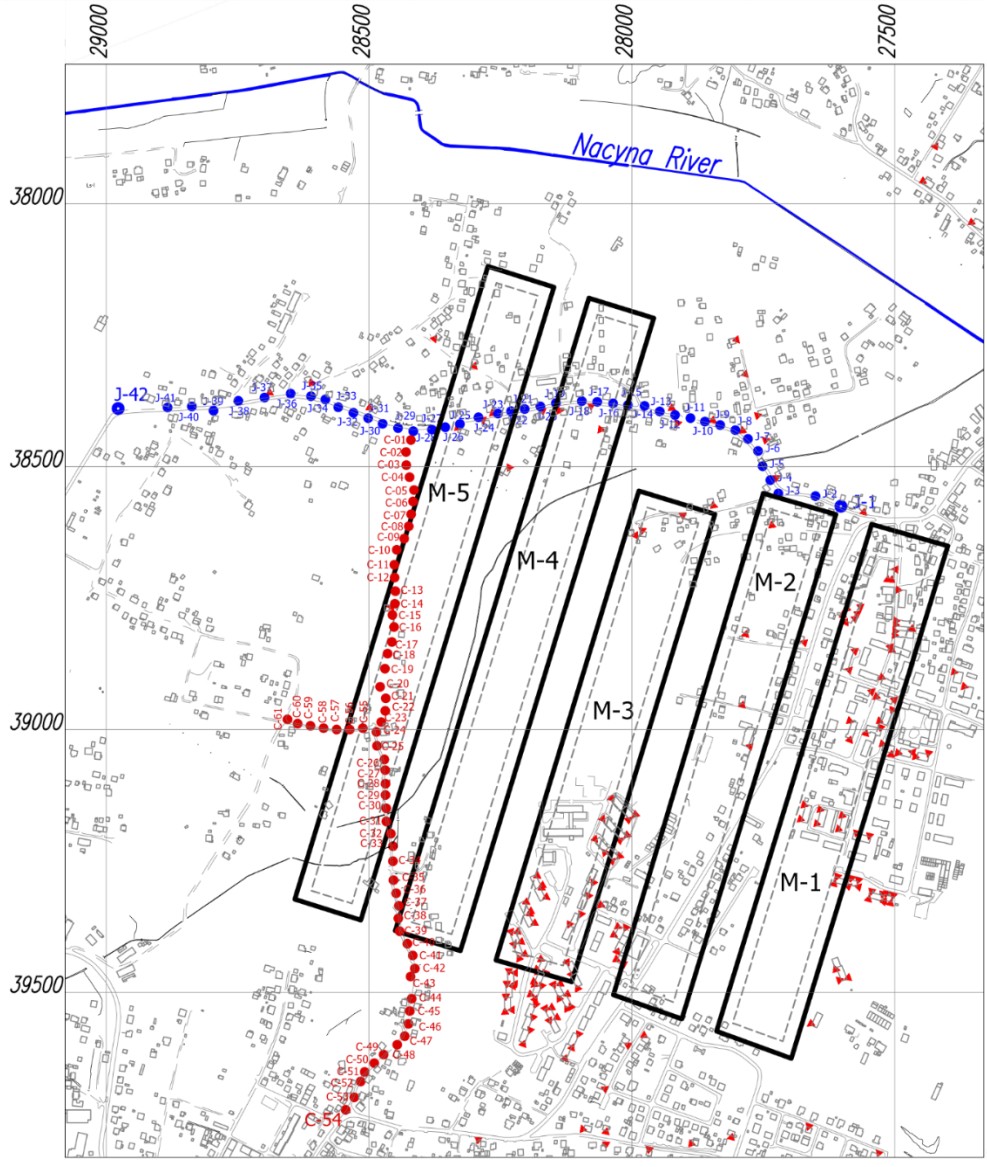

**Figure 1.** Sketch of planned longwalls contours in the seam $712/_{1-2}$.

The selection of longwall length and pillar width aimed to:

- Ensure the adequate load-bearing capacity of pillars;
- Due to the threat of rock bursts, minimize the stress influences of a previous longwall panel's abandoned workings edges on workings in the currently exploited longwall;
- Minimize the mining area deformations and simultaneously minimize losses in the deposit.

The load-bearing capacity of the coal pillars and the assessment of the state of stress were estimated with the use of the finite element method, while the influences of the planned exploitation were estimated using computer programs with implemented formulas by S. Knothe [2].

Prognostic calculations of terrain deformations due to the exploitation of M5 ÷ M1 longwalls were performed with the assumption of the following values of S. Knothe's theory parameters: $a = 0.8$, $tg\beta = 2.7$ and $d = 25$ m (the exploitation rim was set down in a geometric manner). The contours of longwalls, taking into account the exploitation rim, are marked by a dashed line in Figure 1. The parameters values were determined on the basis of measurements of subsidences for the observation line points along Górnośląska Street in Niedobczyce, showing the influences of previously exploited seams $624/_1$, $626/_1$, $702/_1$ and $707/_2$. This forecast has shown that the implementation of the presented exploitation project would result in the occurrence of subsidences reaching the depth of up to 1.15 m and maximum horizontal deformations not exceeding the limit of 3.0 mm/m [3].

The preliminary forecast of the M-5 ÷ M-1 longwalls' influence in seam $712/_{1-2}$ concerned the incomplete subsidence troughs caused by the exploitation of deep and very narrow longwall panels.

Such forecasts are characterized by a high level of uncertainty. Therefore, the results of this forecast were additionally verified by comparing them with subsidences measured on *J* line and *C* line in Niedobczyce (Figure 1), obtained after extracting the M-5 and M-4 longwalls and determined by means of numerical modeling with the finite difference method, using the Fast Lagrangian Analysis of Continua (FLAC) computer program.

At the same time, this provided an opportunity to check the applicability of numerical modeling using the finite elements method to describe deformations in the terrain surface in the case of experimental exploitation with narrow belts, leaving the pillars of coal.

## 2. Geological and Mining Conditions in the Area of Exploitation Conducted in Seam 712/1-2

In the past, an intensive mining exploitation, mainly of a roof caving type, was conducted under the Niedobczyce district of the city of Rybnik. In the years 1813–2008 it covered seams from 602 to $626/_{1-2}$, $703/_{1-2}$ and $707/_2$.

Since 2012, the mine has been carrying out experimental mining operations with the caving of roof rocks in seam $712/_{1-2}$. This is the last seam in this area that can be extracted. According to the project, in the years 2012–2014, M-5 and M-4 longwalls, with lengths of approximately 130 m and panel lengths of 1250 m, were extracted, and by the end of 2018 the same will happen to M-3, M-2 and M-1 longwalls, with lengths of approximately 150 m and panel lengths equal to 930 m (longwall M-3) and 1020 m (longwalls M-1 and M-2).

The depth of exploitation varies from 1120 m in the area of the M-5 longwall up to 960 m in the area of planned M-1 longwall (mean average, 1040 m). The height of the longwalls is 3.0 m.

Using the results of research on the influence of the order and direction of exploitation on the shape of subsidence troughs (larger deformations in the beginning of the exploitation area), it was assumed that the order of extracting longwalls should be inverse to their numbering [4]. This entailed beginning the exploitation in a sparsely built-up area of the M-5 longwall and ending it in the most protected area of the historic housing estate on Andersa Street and Barbary Street, situated in the area of the M-1 longwall influences.

In the area that the conducted exploitation of seam $712/_{1-2}$ influences, the geological conditions are typical for the Upper Silesian Industrial Region. Overburdened layers with a thickness of about 50 m are made of quaternary and tertiary formations. Quaternary formations are developed in the form of sandy clays and loess-like dusts with the thickness of 2–6 m, lined by gravels and sands of various grain sizes with a thickness of up to 15 m. The tertiary is developed in the form of plastic, water-impermeable gray-green loams.

The carboniferous consists of "porębskie" and "jaklowieckie" layers. In terms of lithology, carboniferous layers are developed mainly in the form of alternating layers of loam and sand slates and sandstones, interbedded by coal seams. Carboniferous deposits fall from east to west with an

inclination of around 4–8°, whereas their extent is approximate to the north–south direction with a slight deviation to the east.

## 3. The Results of Geodetic Observations

The influence of the M-5 and M-4 longwall exploitation in seam $712/_{1-2}$ were observed in an altitudinal and linear manner on two approximately perpendicular observation lines along Janasa Street, *J* line, and along Chodkiewicza Street, *C* line (Figure 1).

The *J* line consists of 42 ground points placed every 36 m on average, and the *C* line of 54 ground points stabilized every 26 m on average.

During the exploitation of the M-4 and M-5 longwalls in seam $712/_{1-2}$ on *J* line, seven measurement cycles were carried out in semi-annual periods (initial cycle, 10 November 2011 and the seventh cycle, 28 November 2014), while on the *C* line, six observation cycles were carried out (the first cycle, 11 September 2012 and the sixth cycle, 27 November 2014) also in six-month periods.

The results of measurements made on the *J* and *C* lines in the last measurement cycle show the deformations that appeared after extracting the M-5 and M-4 longwalls. The maximum subsidence observed at that time was 0.67 m on *J* line (point 24), and 0.60 m on *C* line (point 19).

Horizontal deformations determined in subsequent measurement cycles were on the *J* line within the range of −1.93 mm/m (Section 24–25) to +1.68 mm/m (Section 17–18), while on the *C* line they ranged from −2.7 mm/m (Section 8–9) to +2.80 mm/m (Section 9–10). The determined values of horizontal deformations from numerical calculations do not exceed the permissible limits in this region ($\varepsilon < 3.0$ mm/m).

## 4. Simulation of Rock Mass Deformations with the Use of the Finite Difference Method (Rock Mass Is Treated as a Layered, Continuous, Transversely Isotropic Medium)

Numerical modeling of mining terrain deformations carried out for the purpose of this work was conducted using the FLAC computer program by the Itasca Consulting Group, based on the finite difference method developed in 1986 by Dr. Peter Cundall and his associates from the University of Minnesota and the Itasca Consulting Group [5].

On the basis of the geological and mining conditions described above, a numerical model of rock mass and planned exploitation in seam $712/_{1-2}$ was built. This model presents a spatial body with the base of 2600 m × 2600 m, and height of 1600 m (Figure 2). Locations of measurement points situated on lines stabilized along *J* line and *C* line were mapped on the surface of the model.

Seam $712/_{1-2}$, with a thickness of 3 m, was mapped on an average depth of approximately 1040 m. Above this seam, 20 subsequent layers describing the carboniferous rock mass and one layer constituting a substitute rock mass of overburden layers were modeled. The floor of seam has been described using six layers.

All layers of the rock mass model were described by the elastic transversely isotropic medium [6,7]. Only in the case of the coal seam was an elasto-plastic model assumed, which was described with the Coulomb–Mohr criterion. Such an assumption will enable mapping the partial destruction of the coal belts separating the designed longwalls. The strain parameters adopted for the calculations were determined on the basis of the works [6,8]. They allow mapping the subsidence trough with the slope coefficient's value $A_T = 2.7$ (Table 1). In the case of the transversally isotropic model, different slopes of the subsidence trough profile are obtained by varying the deformation parameters between the isotropy planes and the direction perpendicular to them. The scheme of the selection of deformation parameters in order to obtain a trough with a determined inclination value is presented in the work of [6]. The value of the $A_T$ indicator in the case of a complete subsidence trough corresponds with the value of parameter $tg\beta$ occurring in S. Knothe's theory, determined on the basis of maximum slopes measurement.

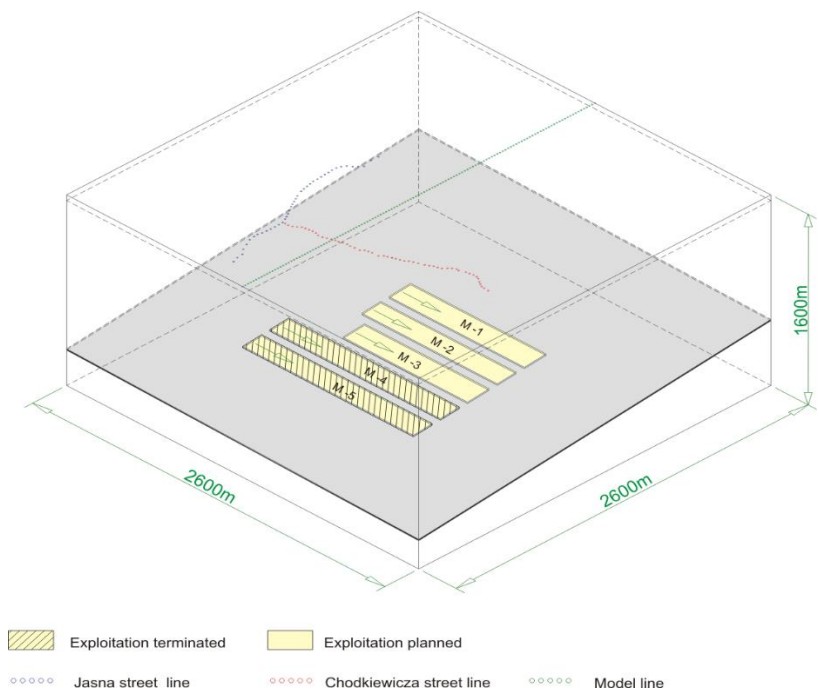

**Figure 2.** Diagram of the numerical calculation model.

**Table 1.** Layers' parameters for the ubiquitous joint model selected for calculations.

| Parameter | Unit | Carboniferous Rock Mass | Overburden |
|---|---|---|---|
| Longitudinal modulus of elasticity ($E_1$) | [MPa] | 10,000 | 8 |
| Longitudinal modulus of elasticity ($E_3$) | [MPa] | 10,000 | 100 |
| Poisson's ratio ($\nu_{12}$) | [-] | 0.12 | 0.3 |
| Poisson's ratio ($\nu_{13}$) | [-] | 0.12 | 0.3 |
| Shear modulus ($G_{13}$) | [MPa] | 220 | 38.5 |

In order to obtain a correct description of the horizontal deformations of the mining terrain, contact layers were introduced between the distinguished layers, this way simulating an inter-layer connection [8–10]. The parameters of the inter-layer connections are shown in Table 2.

**Table 2.** Mechanical parameters of the inter-layer connections.

| Parameter | Unit | Value |
|---|---|---|
| Normal stiffness coefficient ($K_n$) | [MPa/m] | 50 |
| Tangential stiffness coefficient ($K_s$) | [MPa/m] | 20 |
| Friction angle ($\varphi$) | [degrees] | 28 |
| Consistency (c) | [MPa] | 0.9 |
| Tensile strength ($R_r$) | [MPa] | 0.01 |

In the pre-plastic state, contact elements behave identically to elements describing rock layers. After transition into a plastic state, they have the possibility of much larger distortions from the traditional zones of the finite difference mesh discretizing the continuous medium.

The body of rock mass has been divided by a mesh, creating cuboidal zones. At the same time, it was assumed that nodal points—located on the extreme side planes and on the basis of the model —can only move within the limits of these planes. Other nodes of the model can move freely in any

direction. The value of primary stresses in the rock mass was determined assuming that they originate only from gravitational forces [11–13].

The simulation of exploitation consisted of the cyclical removal of individual zones of the finite difference mesh in an order corresponding to the extraction of the longwalls. Gradually, along with the development of the post-mining void in seam $712/_{1-2}$, contact elements were implemented between the roof and floor layer of seam in the form of a separation plane which prevents mutual penetration of roof and floor of seam. Such a way of simulating the mining exploitation eliminates the necessity of implementing additional parameters attributed to caving zones. This way of modeling a complete subsidence trough for exploitations of a sufficiently large size (compared to the depth of exploitation) would make the settlement coefficient *a* approach the value of *a* = 1.0.

## 5. Verification of the Calculation Model Based on the Results of Subsidences Measurements after the Exploitation of M-5 and M-4 Longwalls

The results of computer simulation carried out using the FLAC program were checked by comparing the subsidences obtained from numerical modeling with subsidences measured after extracting the M-5 and M-4 longwalls. The results of this comparison are shown in Figure 3—the results of the measurements and subsidences of 75 points of the *J* and *C* lines were collected together. It was assumed that between the analyzed subsidences there should exist a regressive relationship:

$$W_{mo} = x \cdot W_{measured} + \Delta W + e \tag{1}$$

where $W_{mo}$ is the subsidences obtained from numerical modeling, $W_{measured}$ is the measured subsidences, $\Delta W$ is the increase of subsidences, $e$ is the mean error of the model, and $x$ is the parameter of the regression curve.

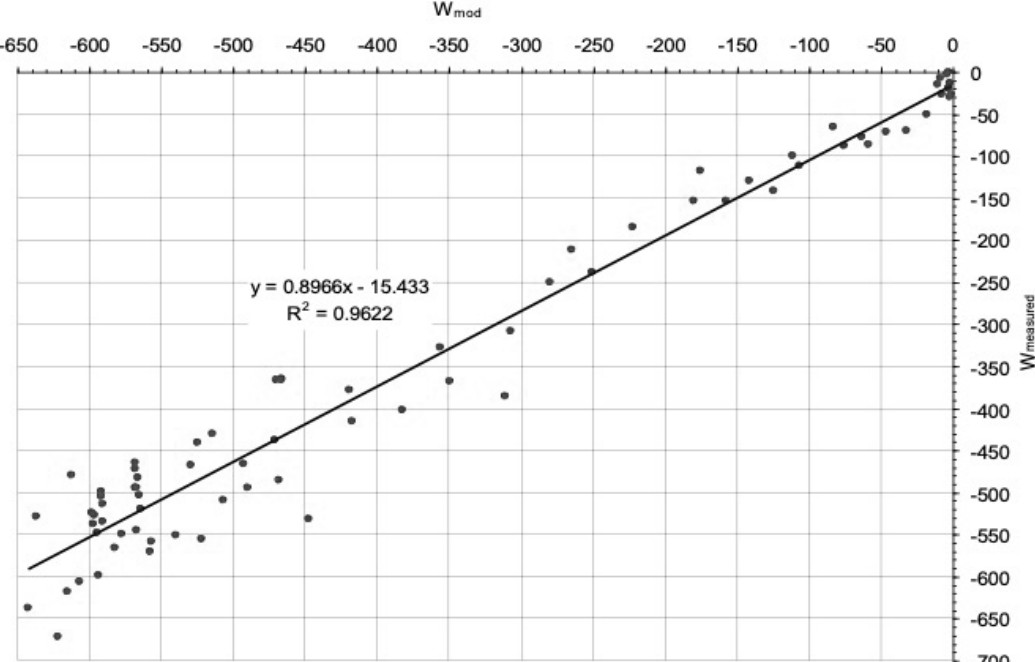

**Figure 3.** Results of comparing the model subsidences and geodetic measurements of 75 points on the *J* and *C* lines.

This linear relationship leads to the following conclusion:

$$W_{mod} = 0.8966 \cdot W_{measured} - 15.433 \tag{2}$$

This relationship is characterized by a very high coefficient of determination $R^2 = 0.96$. Its mean error reaches the value of 52.7 mm, and the corresponding coefficient of variation is $M_w = 100\%$ * 52.7 mm/592 mm = 8.9%.

Some permanent subsidence of about 15 mm is present here, which can be attributed to the remote influences caused by the dehydration of rock mass and activations of abandoned workings.

The determined value of the regression coefficient $a = 0.8966$ can be approximated to some extent as the settlement coefficient, appropriate for the conducted numerical calculations. Using relationship (1), the subsidences obtained as a result of numerical modeling with the use of the FLAC program were recalculated and are shown in the form of graphs in Figures 4–6.

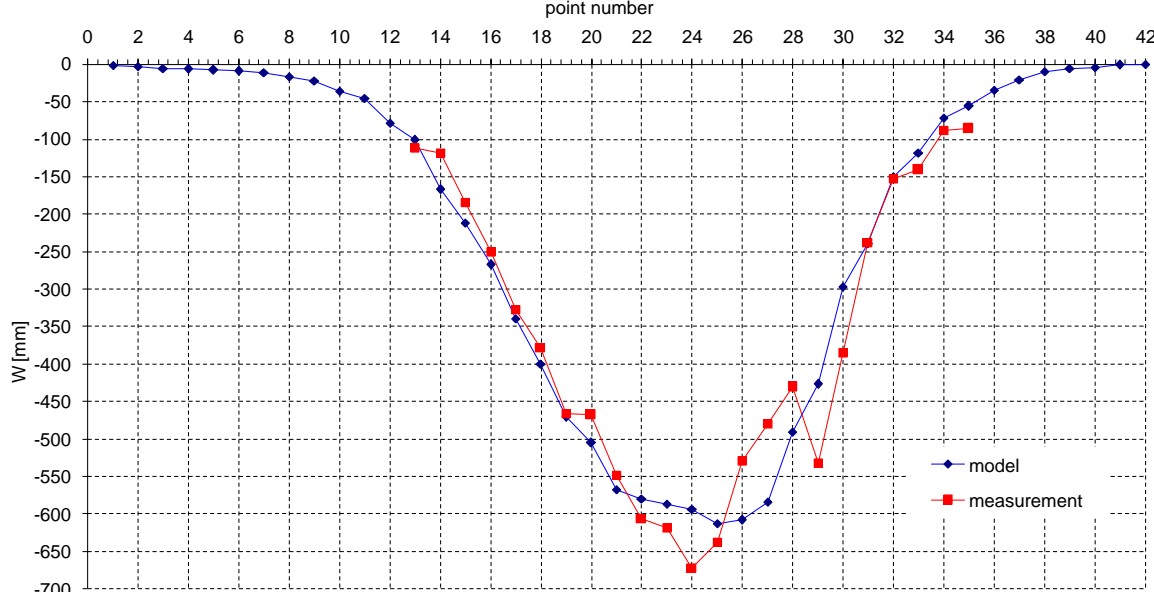

**Figure 4.** Comparison of the theoretical trough determined on the basis of numerical modeling with the results of geodetic measurements—the *J* line.

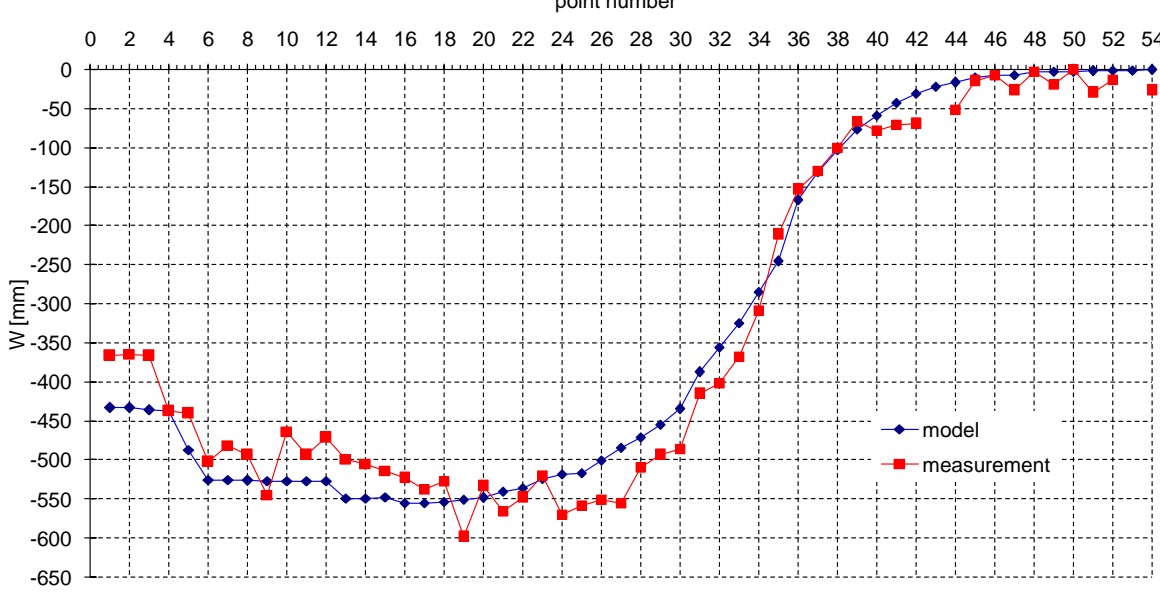

**Figure 5.** Comparison of the theoretical trough determined on the basis of numerical modeling with the results of geodetic measurements—the *C* line.

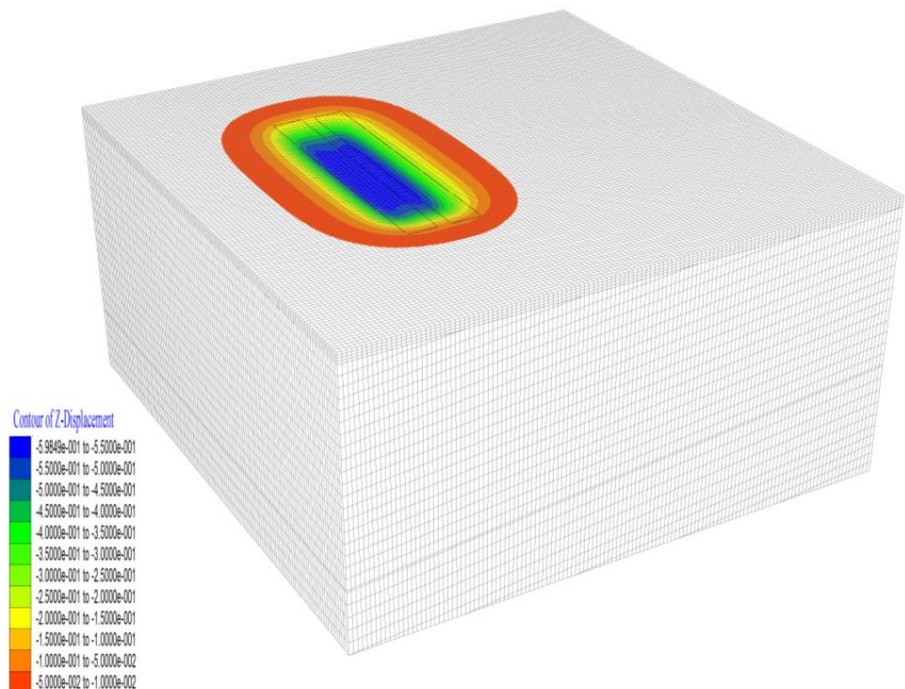

**Figure 6.** Map of subsidences determined after extracting the M-5 and M-4 longwalls.

Of note is the very good mapping of the subsidence trough, both in terms of the values of the subsidences and the maximum slopes. This indicates a good mapping of the mining terrain subsidences by a transversely isotropic rock mass model described by parameters (Table 1) determined on the basis of the results of research for $A_T = 2{,}7$. Similar results were also obtained in the case of analysis of surveying measurements presented in the work of Białek et al. [14].

## 6. The Effects of Computer Simulations of Mining Terrain Deformations as a Result of the Planned Exploitation of Seam 712/$_{1-2}$

The numerical model of describing deformations verified by the results of geodetic measurements of subsidences was a basis for a multi-stage numerical simulation of the mining exploitation of the M-5 and M-4, M-3, M-2 and M-1 longwalls. The maximum values of the mining terrain deformation indicators obtained for the model line (located transversely to the longwalls' panel lengths) are presented in Table 3, while the course of their variability is illustrated in Figures 7–10.

**Table 3.** Deformation indicators determined from the model after extracting subsequent longwalls.

| Longwall | $W_{max}$ [mm] | Static Edge | | | Moving Edge | | |
|---|---|---|---|---|---|---|---|
| | | $\varepsilon+$ [mm/m] | $\varepsilon^-$ [mm/m] | $T_{max}$ [mm/m] | $\varepsilon+$ [mm/m] | $\varepsilon^-$ [mm/m] | $T_{max}$ [mm/m] |
| M-5, M-4 | 600 | 1.022 | −1.726 | 3.18 | 1.001 | −1.737 | 3.25 |
| M-5, M-4, M-3 | 1091 | 1.546 | −2.469 | 4.66 | 1.395 | −2.068 | 4.02 |
| M-5, M-4, M-3, M-2 | 1337 | 1.561 | −2.757 | 4.88 | 1.548 | −1.156 | 4.24 |
| M-5, M-4, M-3, M-2, M-1 | 1478 | 1.562 | −2.925 | 4.89 | 1.548 | −1.768 | 4.70 |

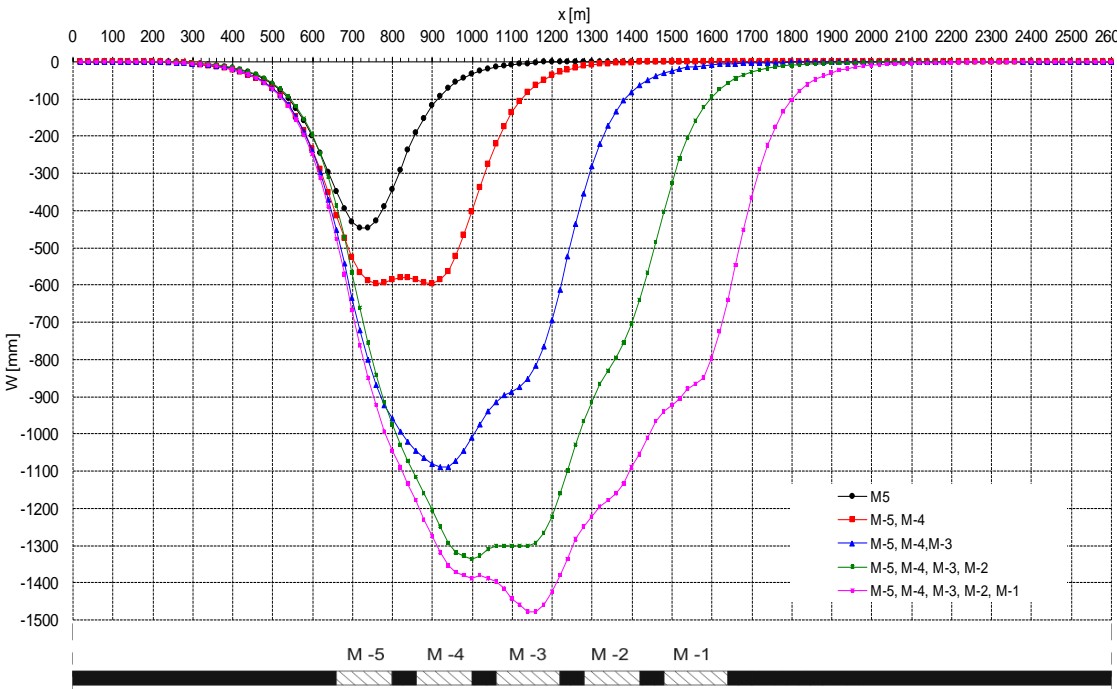

**Figure 7.** Subsidences determined on the basis of numerical modeling on the model line after extracting subsequent longwalls.

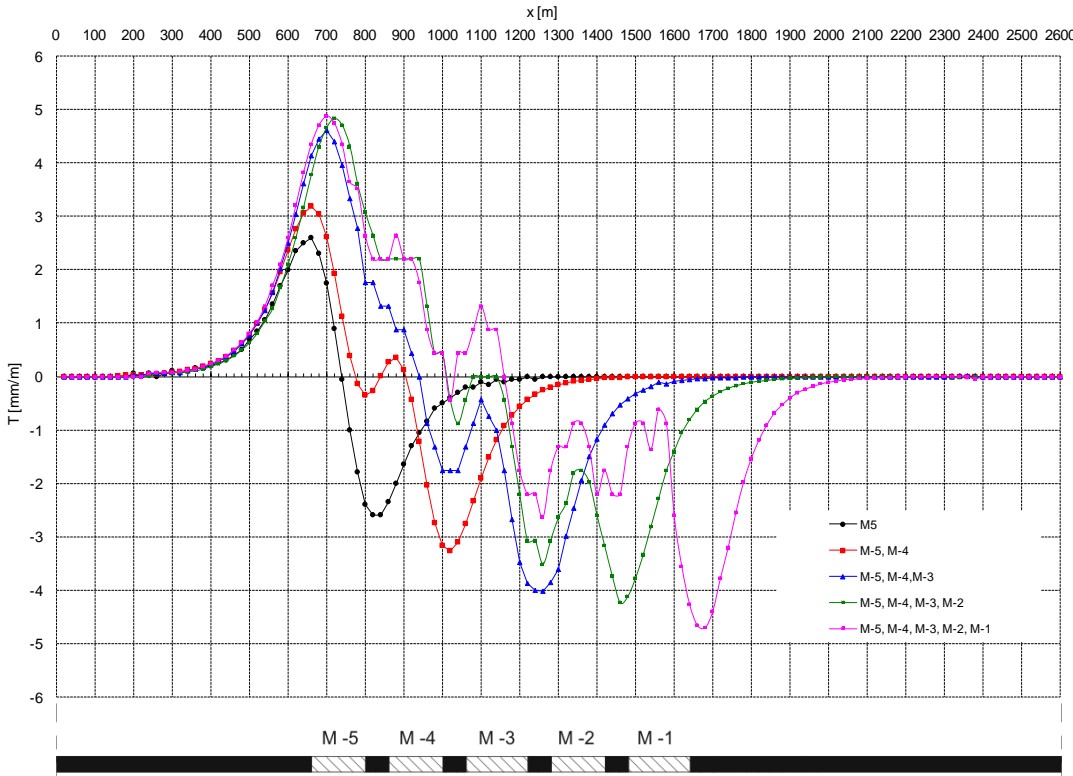

**Figure 8.** Slopes determined on the basis of numerical modeling on the model line after extracting subsequent longwalls.

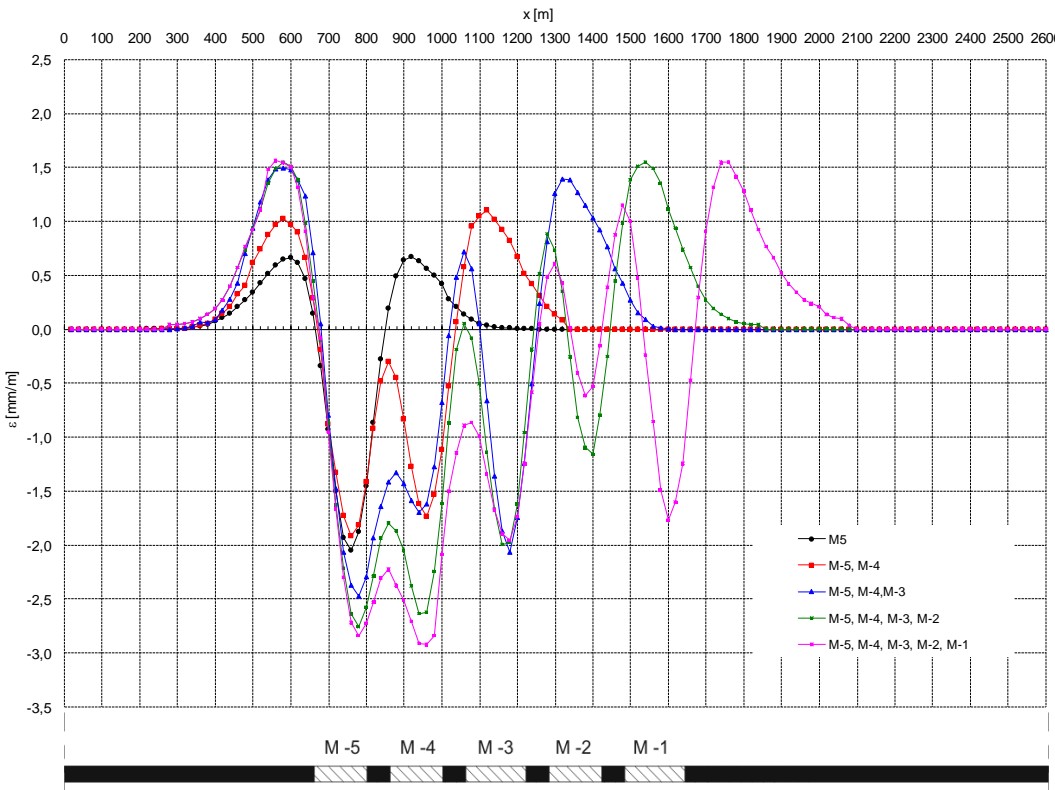

**Figure 9.** Horizontal deformations ε determined on the basis of numerical modeling on the model line after extracting subsequent longwalls.

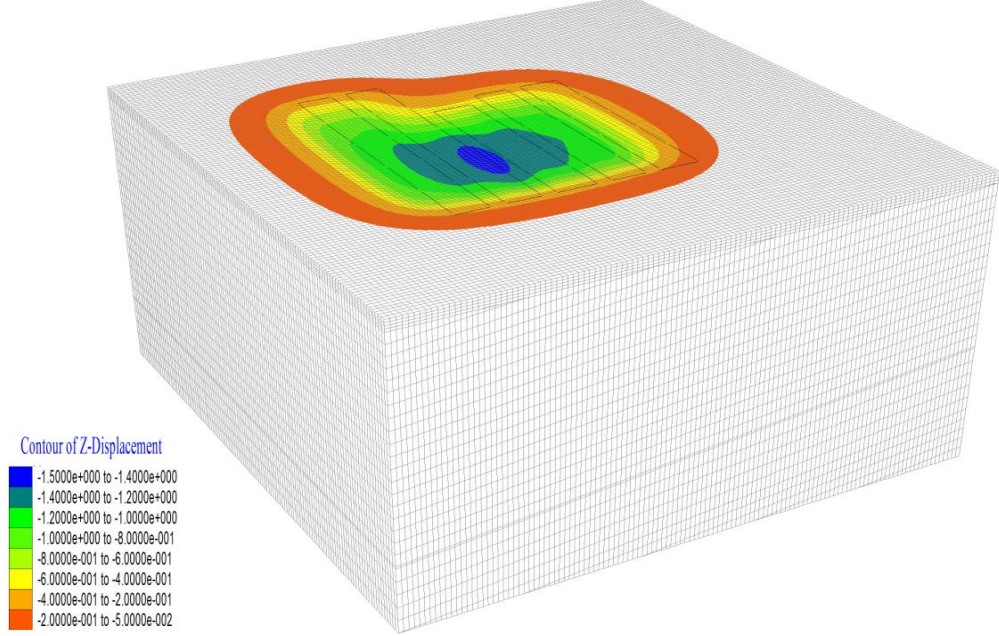

**Figure 10.** Map of subsidence determined after extracting all longwalls.

The results of numerical calculations lead to the following conclusions.

The maximum value of subsidence (along the calculation line) after extracting all planned longwalls is approximately 1478 mm (Figure 7). The maximum increase of subsidences caused by the exploitation of longwalls is $\Delta W_{M-5}$ and $\Delta W_{M-4}$ = 600 mm, $\Delta W_{M-3}$ = 491 mm, $\Delta W_{M-2}$ = 246 mm and $\Delta W_{M-1}$ = 141 mm, respectively. Small increases in subsidences determined for the M-2 and M-1 longwalls indicate that subsidences resulting from the extraction of all longwalls are close to the

maximum subsidences of the complete trough, with partial exploitation designed this way. The final results were partially confirmed by geodetic measurements, which were made in July 2018. On the Chodkiewicza Street line, the maximum subsidence amounted to approximately 1490 mm.

The analogous value of the maximum vertical displacements obtained from the numerical calculations reached the value of about 1385 mm.

After extracting the M-5 and M-4 longwalls, the maximum slopes of both wings of the analyzed troughs are practically the same, which is the result of the same geomechanical conditions during the formation of both sides of the trough. Along with the development of mining exploitation in subsequent plots, there appears a slight difference in the slopes of the subsidence trough over the static edge of exploitation (M-5 and M-4 longwalls) and the maximum slope of the trough's wing over the edge, changing its location (Figure 8). The slopes over the moving edge are slightly larger than the slopes above the static edge. Such distribution of maximum slopes is related to various geomechanical and mining conditions (the length of longwalls) during the formation of the trough's sides. However, for each exploitation stage, the maximum slopes will not exceed 5.0 mm/m [4,15].

The results of the numerical modeling in terms of horizontal deformations (Figure 9) indicate that the extreme values of tensile strains will not exceed the value of 3.0 mm/m in any stage of exploitation. This concerns both the beginning of the exploitation edge (M-5 longwall area) and the edge changing its location. After extracting the first two longwalls M-5 and M-4, the maximum tensile strains of both wings are similar and amount to 1.05 mm/m (static edge) and 1.15 mm/m (moving edge). With the development of mining exploitation in subsequent plots, the values of deformations gradually increase. There is a slight difference in deformations over the static edge of the exploitation (M-5 and M-4 longwalls area) and over the edge changing its location. Deformations over the moving edge are smaller than the corresponding deformation values over the static edge. The presented results show that horizontal deformations will not exceed the assumed values for any of planned mining exploitation stages. The extreme strain values determined after extracting all planned longwalls are 1.5 mm/m for stretches and −2.85 mm/m for compressions, respectively. Such small values of horizontal deformations also do not cause the risk of discontinuous deformations. This is confirmed by the results of work carried out by P. Strzałkowski [16]; P. Strzałkowski and K. Tomiczek [17] and A. Kowalski [18].

Figure 10 presents a map of a finally developed subsidence trough after extracting all longwalls.

On the surface, near the extracted longwalls, non-cyclical leveling measurements of dispersed points were carried out using the GNSS RTN (Global Navigational Satellite System; Real Time Kinematic) method. The location of the measuring points is marked with triangles in Figure 1. The last measurements were made in July 2018. Despite the accuracy, the limitations of the method used and the location of the measuring points in comparison to the location of the calculation points, it should be stated that the results of the observations coincide with the results of the numerical calculations. This applies to both qualitative and quantitative compliance. Unfortunately, the location of these points precludes, in principle, a full comparison with respect to the numerical model.

## 7. Conclusions

This paper presents the results of modeling the surface deformation of the mining terrain caused by the conducted partial exploitation. The model of rock mass was based on both the elastic transversally isotropic medium and the isotropic medium that takes into account the plastic features of the extracted layers. The computer simulation was carried out on the basis of actual exploitation in seam $712/_{1-2}$ of the Marcel Coal Mine, with the use of the results of geodetic measurements showing deformations in the initial stage of seam's exploitation. Based on the results of the numerical modeling, the following final conclusions were formulated:

1. Based on the comparative analysis of the results of the numerical modeling and the geodetic measurements conducted during the exploitation of the M-5 and M-4 longwalls, it can be concluded that the strength and strain parameters of the model have been properly adjusted and the numerical

model can be used to assess the influence of planned exploitation on deformations of the mining terrain surface.

2. The presented results of the numerical modeling explicitly indicate that for the assumed mining and exploitational conditions the extraction of all plots will lead to the emergence of a subsidence trough on the surface, in which the maximum subsidences will be located over the central part of the panel of the M-3 longwall and will amount to approximately 1480 mm. The results were confirmed by measurements, which were made in July 2018. The subsidence of the terrain will result in changes to its slopes as well as changes in horizontal deformations.

3. As shown by the results of the presented computer simulations, the use of the partial exploitation of the coal seam may be an effective way of conducting exploitation under built-up areas. The prognostic calculations show that the value of deformation indicators never exceeds the assumed limit values. This also applies to the Nacyna River.

Based on the results of calculations performed as part of this work, it can be concluded that partial exploitation can be an element of the rational management of deposits, especially when the only alternative is completely abandoning of exploitation in a given area.

**Author Contributions:** Software, M.W.; formal analysis, J.B.; supervision, R.M.; writing—review and editing, P.S. All authors have read and agreed to the published version of the manuscript.

**Funding:** This research received no external funding.

**Acknowledgments:** The article presented above is based on the results of the prognosis of the impact of underground mining operations in the seam 712/1-2 made as part of the work ordered by KWK Marcel.

**Conflicts of Interest:** The authors declare that they have no conflict of interest.

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
