# Peer review of "Deformations of Mining Terrain Caused by the Partial Exploitation in the Aspect of Measurements and Numerical Modeling"

_sustainability, doi:10.3390/su12125072_

Round 1

Author Response

Comment on the reviewer's 1 suggestions and questions

Chapter.1.

Quotation: According to the preliminary forecasts [1] the planned roof caving exploitation of the 712/1-2 seam carried out to the full extent (without leaving belts of coal solid) would cause the emergence of III category deformations. This fact incurred a strong resistance of the residents which led to the decision of abandoning the mining activities in this area. The only socially acceptable solution was the exploitation causing deformations of the II category with subsidences in the area of Nacyna river limited to 0.1 m.

- For foreign readers, categories II and III should be defined. These are the categories used in Poland.

- The Nacyna River should be placed in Figure 1.

Answer: Both the Nacyna River and the provision of limit values for deformation rates have been included in the text.

Prognostic calculations of terrain deformations due to the exploitation of M5÷M1 longwalls were performed with the assumption of the following values of S. Knothe’s theory parameters a = 0.8, tgβ = 2.7 and d = 25 m (the exploitation rim was set down in a geometric manner). The description in brackets does not correctly explain the concept of exploitation rim. Can be shown possibly in Figure 1.

Answer: Operating contours including the operational rim were marked by dashed line in Fig. 1.

This forecast has shown that the implementation of the presented exploitation project would result in the occurrence of subsidences reaching the depth of up to 1,15 m and maximum horizontal deformations not exceeding the limit of category II [3]. With reference to the calculations presented in Figure 7, the forecast, i.e. wmax (according to Knothe) is 1.15 m and the FLAC model is 1.5 m. An important difference.

Answer: The forecast of the impact of the exploitation on the land surface was made on the basis of the EDN program of prof. Białek. Determined values of subsidences with this program (including the operational rim) are max 1.15 m. From the numerical calculations made with the FLAC program, the max value of subsidence is approximately 1.5 m. It is worth noting here that the parameter of the operational edge is used only in the geometric-integral theory. In the FLAC program it is not necessary to include the operational edge (d parameter). Frequently observed point shift of 0.5wmax (max subsidence) towards selected space is the result of direct numerical calculations. This has been pointed out many times in our publications [including in publication No. 8].

Therefore, the results of this forecast were additionally verified by comparing them with subsidences measured on Janasa Street and Chodkiewicza Street in Niedobczyce (Figure 1) obtained after extracting the M-5 and M-4 longwalls and determined by means of numerical modeling with the finite difference method, using the FLAC computer program. I suggest describing the lines through Line 1, Line 2 (or J and C), not by street names. Later in the article, also change the name of the line.

Answer: Taking into account the reviewer's suggestion, the names of the measuring lines were changed.

Chapter 2

Using the results of research on the influence of the order and direction of exploitation on the shape of subsidence troughs (larger deformations in the beginning of exploitation area) it was assumed that the order of extracting longwalls should be inverse to their numbering [4]. This means beginning the exploitation in a sparsely built-up area of the M-1 longwall and ending it in the most protected area of the historic housing estate on Andersa Street and Barbary Street, situated in the area of the M-1 longwall influences.

If I understood correctly should: … a sparsely built-up area of the M-5 longwall…. (not M-1)

In figure 1, show the boundaries of the historical housing estate. Street names are irrelevant to a foreign reader. In addition, these streets are not on the map (Fig.1)

Answer: The change has been taken into account in the text (chapter 2).

Chapter 3

The maximum values of these horizontal deformations fall into category II of the mining terrain Proposition: Put a bracket at the end of the sentence (ɛ<3,0mm/m)

Answer: The change has been taken into account in the text (chapter 3).

Chapter 4

They allow mapping the subsidence trough with the slope coefficient’s value AT = 2.7 (Table 1) This value is not in Table 1.There is no explanation in the article as to how this parameter was used.

Answer: In the case of the transversally isotropic model, different slopes of the sinking basin profile are obtained by varying the deformation parameters between the isotropy planes and the direction perpendicular to them. The scheme of selection of deformation parameters in order to obtain a subsidence trough with any inclination value is presented in [6]. The explanation has been included in the text (chapter 4).

Chapter 5

It was assumed that between the analyzed subsidences there should exist a regressive relationship.

ambiguous symbols: e – mean error of the model. a - the regression coefficient

Chapter 6, Fig 9 e - horizontal deformations In Chapter 4 a - settlement coefficient (exploitation coefficient)

Answer: The changes have been included in the text.

Chapter 5

Fig.4 - Why the measurement results are not shown on points 1-12 and 36-42. No correct match on impact range? Requires clarification. Legend for Fig. 5 and Fig.10 (Chapter 6 ) illegible. Change the labels in the legend (larger font, subsidence in meters to 0.01)

Answer: In Figure 4, some points describing the so-called far operational impacts were removed. The values of the subsidences at these points were difficult to associate with the current mining and geological situation in this region. Therefore, the reasons for the measured subsidences at such points could not be identified. As an explanation, we also attach the full course of the measured subsidences on the J line.

Chapter 6

The maximum values of mining terrain deformation indicators obtained for the model line (located t ransversely to the longwalls panel lengths) are presented in Table 3, while the course of their variability is illustrated in Figures 7-10.

Draw the course of this line in the drawing, because we do not know too much about its course and it is impossible to estimate the occurrence of extreme deformation and displacement values.

Answer: The course of the model line is shown in Figure 2.

On the surface of the area, near the extracted longwalls, non-cyclical leveling measurements of dispersed points were carried out using the GNSS RTN method. The last measurements were made in July 2018. Despite the accuracy limitations of the method used and the location of the measuring points in compare to loacation (Should be: location)of calculation poins (Should be: points), it should be stated that the results of observations coincide with the results of numerical calculations. This applies to both qualitative and quantitative compliance. Why the measurement results are not shown? No distribution of these points, charts etc. The most interesting stage from the point of view of practitioners, i.e. verification of a given model/calculation method, was omitted. And it is described in only one sentence. This paragraph would be a very good summary of the whole article

Answer: The location of the dispersed points is shown in Figure 1. On their basis, it can only be compared the values of the subsidences in these points. The location of these points precludes, in principle, a full comparison with respect to the numerical model.

Reviewer 2 Report

In the Introduction, it states that "the only socially acceptable solution was the exploitation causing deformations of the II category with subsidences in the area of Nacyna river limited to 0.1 m". However the remainder of the paper presents measured subsidence data up to 0.6 m and higher levels predicted with future mining. I am unsure as to the significance and relevance of this statement in the introduction as I was expecting far lower measured values than was the case.

I have struggled to get Figures 4 and 5 to align with the measuring points indicated in Figure 1, in particular the point at which the Janasa Street (JS) Line and Chodkiewicza Street (CS) Line come together at points J28 and C01 in Figure 1. Point J28 has measured subsidence in the order of 0.5 m, yet point C01 is essentially zero, but is very close to point J28. This doesn't seem to make sense and I am wondering whether the numbering of measurement points along the CS Line is back to front - i.e. point C54 should be at the intersection with the JS Line, which would then in fact make perfect sense based on the measured subsidence variations along the CS Line. 

It would be helpful, as you have done in Figure 7, that Figures 4 and 5 contain the location of extracted void and solid coal as reference points rather than the reader needing to keep referring back to Figure 1. The importance of this can be demonstrated by noting that following the extraction of M5 and M4, the centre of the measured subsidence trough (point J24 in Figure 4) should presumably be above or at least close to the centre of the M5 to M4 coal pillar, as it is for the modelled subsidence profile in Figure 7 for the specific case where M5 and M4 are extracted. However in Figure 1, point J24 is directly above the centre of the M5 extracted area, not the M4 to M5 pillar. I am interested as to why this discrepancy is evident between the measured data and modelled/predicted data in terms of the lateral shift in subsidence trough location relative to the underlying mining layout.       

It would also be valuable to include a discussion on subsidence mechanisms, in particular sag subsidence as opposed to that due to coal compression. At 1000 m cover depth, the amount of sag subsidence at surface across a 130 m wide longwall panel with a width to depth ratio (W/D) of only 0.13, is likely to be small compared to that due to coal compression around the panel. This is clearly the case from the modelled data in Figure 7 for M5 extraction only whereby in the area where M5 has been extracted, the differential subsidence actoss the panel is only about 100 mm out of a total of about 430 mm, the remainder then being linked to coal compression, which is not unsurprising given the cover depth involved. The significance of coal compression in the subsidence profile across several panels is then demonstrated in Figure 4 and Figure 7 whereby the influence of chain pillars between panels in the surface subsidence profile is almost nothing (both modelled and measured), this indicating that surface subsidence is being controlled by significant chain pillar compression effects rather than overburden sag effects above the extracted areas, this being identical to that measured in the Southern Coalfield of NSW in Australia for example at cover depths in the order of 500 m.

I have no disagreement that the numerical model closely matches the measured profiles, but would argue that the key aspect of the model is unlikely to be the rock mass properties being correctly adjusted, but the use of an elasto-plastic behaviour model for coal, thereby allowing the coal to compress significantly once it is over-stressed. It would be an interesting exercise to turn this aspect off in the model so that pillars are elastic only, as I am guessing that the predicted surface subsidence profile would reduce very dramatically as a direct consequence.

Overall this is a valuable and interesting piece of work, in particular its use in justifying coal extraction whilst also minimising surface impacts in residential areas. However, there is some confusion in the presentation of the measurement data (or I have missed something in my review) and the authors would add significant value to the paper by addressing the significance and relevance of different subsidence forming mechanisms rather than solely focusing on confirming the suitability of the numerical model assumptions for the rock mass.                        

Author Response

(The authors gave the same response as above.)
